# Roll-to-roll fabrication of touch-responsive cellulose photonic laminates

Hsin-Ling Liang [1], Mélanie M. Bay [2], Roberto Vadrucci [2], Charles H. Barty-King [1], Jialong Peng [3], Jeremy J. Baumberg [3], Michael F.L. De Volder [1] & Silvia Vignolini [2]

Hydroxypropyl-cellulose (HPC), a derivative of naturally abundant cellulose, can self-assemble into helical nanostructures that lead to striking colouration from Bragg reflections. The helical periodicity is very sensitive to pressure, rendering HPC a responsive photonic material. Recent advances in elucidating these HPC mechano-chromic properties have so-far delivered few real-world applications, which require both up-scaling fabrication and digital translation of their colour changes. Here we present roll-to-roll manufactured metre-scale HPC laminates using continuous coating and encapsulation. We quantify the pressure response of the encapsulated HPC using optical analyses of the pressure-induced hue change as perceived by the human eye and digital imaging. Finally, we show the ability to capture real-time pressure distributions and temporal evolution of a human foot-print on our HPC laminates. This is the first demonstration of a large area and cost-effective method for fabricating HPC stimuli-responsive photonic films, which can generate pressure maps that can be read out with standard cameras.

[1] NanoManufacturing Group, Department of Engineering, Cambridge University, Cambridge CB3 0FS, UK. [2] Bio-inspired Photonics Group, Department of Chemistry, Cambridge University, Cambridge CB2 1EW, UK. [3] NanoPhotonics Centre, Cavendish Laboratory, Cambridge University, Cambridge CB3 0HE, UK. These authors contributed equally: Hsin-Ling Liang, Mélanie M. Bay. Correspondence and requests for materials should be addressed to M.D.V. (email: mfld2@cam.ac.uk)

With the development of new nanofabrication methods, a variety of new materials have emerged, which show dramatic stimuli-responsive optical behaviours including tuneable colours and scattering properties[1–3]. Directed self-assembly has proven to be a highly promising route to control the organisation of a large variety of nano-photonic building blocks, ranging from colloids[4,5] to liquid crystals[6–8] and block-copolymers[9,10]. However, deterministically controlling the nanoscale material organisation remains challenging, in particular when using scalable manufacturing processes that allow for square meter scale processing. This challenge unfortunately limits the industrial relevance of these exciting emerging materials. High throughput and cost-efficient processing of photonic materials using continuous processes, such as roll-to-roll (R2R) coating are therefore of great importance for their future adoption in commercial devices. R2R coating is a well-established process with applications ranging from commercial printing, to large area electronics[11,12], photovoltaics[13], advanced functional devices such as lighting[14,15], and sensors[16,17], and structural photonic surfaces[18–20]. However, the implementation of R2R on self-organised structures is still in early stages of development, with only very few reports that have been able to overcome some of the challenges of nanoparticle self-assembly on continuous webs[21–25]. It thus remains particularly difficult to unlock the full functionalities of these materials, such as their responsiveness to external stimuli at scale[26–29].

The cellulose derivative hydroxypropyl-cellulose (HPC) is a promising self-assembling photonic material with the advantages of being low-cost (as it is produced on a tonne scale), non-toxic, water-soluble and having been used widely by the medical and food industries as an emulsifier and thickener[30,31]. In a concentrated aqueous solution HPC can form a cholesteric liquid-crystalline mesophase and becomes optically anisotropic[32]. In this case, the HPC molecules self-assemble into helical nanostructures that interact with visible light and produce Bragg-like reflections determined by the helical pitch. The mesophase exhibits vivid colours that can dynamically change through manipulation of pitch, for example, by mechanical compression or expansion. The mechano-chromism of HPC thus provides an effective method to map pressure distributions over large surfaces[26].

We show that low-pressure R2R slot-die coating and lamination allows for the continuous self-assembly of HPC into mechano-chromic packaged films. Further, by controlling the initial HPC to water ratio, we are able to tune the baseline colour of the films to be red, green or blue. In addition to large area continuous coatings, we believe an important step towards the commercial adoption of these materials is an accessible method to translate the mechano-chromic colour changes of the films into a pressure map. Here we show a method that uses simple digital camera images of the HPC films to generate real time pressure maps, which can find applications in sports apparel, medical imaging, strain monitoring and many other areas.

## Results

**R2R processing of HPC laminates.** The HPC lamination process starts with R2R coating of an HPC solution, followed by an edge sealing coating and curing, and a continuous film-to-film lamination to encapsulate the HPC film (Fig. 1a–e). All our coatings are performed on a customised Smartcoater 28 (Coatema Coating Machinery GmbH) R2R tool on 75 μm thick polyethylene terephthalate (PET) films. The selected HPC is of a short chain grade (molecular weight $M_w$ ~33,900) and has a low enough resistance to flow in the mesophase regime, which is suitable for coating processes. The stock HPC solution is prepared by mixing HPC powder with water, followed by de-gassing using high-speed centrifugation.

The characteristic colour of the HPC mesophase can be tuned by changing the HPC concentration (Supplementary Figure 1)[33]. However, the viscosity of the HPC solution increases with concentration of HPC, hence requiring adjustment of the coating parameters for different baseline colours. The low shear (below 1 s$^{-1}$) viscosities of HPC mixtures with initial concentrations in the red, green and blue regions (Fig. 1f) are 80, 147 and 330 Pa s, respectively (Supplementary Figure 2; data acquired at 22 °C at which the coating is performed in this work). While the coating process is adaptable for fabrication of homogeneous films of various colours, we focus on red HPC films as their pressure response spans the full visible spectrum. Water evaporation has to be taken into account during the R2R fabrication process, as the evaporation of water results in a contraction of helical HPC pitch, which again changes the film colour. The average water loss here is 3–4 wt% from the slot-die coating head to the rewinder (see Methods). Here, the starting concertation of the HPC water mesophase is set to 57 wt%, which ensures a final reflection peak in the red part of the spectrum for our material.

Because of its simplicity and ability to coat thick layers, we first attempt a R2R knife-over coating process. We find however that this coating process suffered from 'ribbing' instabilities due to vortex formation in the liquid reservoir[34] when the HPC solution is rolling against the moving substrate (Supplementary Figure 3a, b). This effect is especially pronounced for viscoelastic fluids and often generates 'wet streak' defects (Supplementary Figure 2c). In contrast, the operating window is much wider using slot-die coating[35], which allows for homogeneous deposition of HPC films (see below). Another advantage of slot-die coating of HPC is that the enclosed flow chamber minimises the loss of water during coating, giving better control of the final film colour.

In slot-die coating (Fig. 1b), the interrelated parameters, such as flow rates, slot-die internal geometry, coating gap etc. (see Methods), need to be optimised in order to obtain the desired coating uniformity. Here we target HPC film thicknesses of 800–900 μm and find stable coating conditions using a web speed of 0.1 m min$^{-1}$. A pressure dampener is used to smooth out pressure pulsations from the pump which otherwise results in 'chatter' defects (Supplementary Figure 4). To deposit a viscous fluid such as HPC, it is essential to control the pressure difference between the flow in the pipeline (Hagan–Poiseuille model) and slot-die channel (Couette flow). The slot-die internal pressure drop can be adjusted by tuning the clearance $t$ from the slot (the spacer, or shim, thickness) which scales as $t^{-3}$. Excess or insufficient slot-die internal pressures affect the desired flow rate of the mesophase and hence the coating performance. We implement a $t = 1$ mm clearance slot to give a large enough feed pressure to stabilise the exit meniscus and generate a uniform coating.

After coating the HPC, two stripes of UV-curable pressure-sensitive adhesive are applied adjacent to the edges of HPC coating via 3D-printed nozzle dispensers (Fig. 1c). This adhesive provides edge sealing and acts as a spacer to define the thickness of the laminate. The web is then UV irradiated to partially cross-link the glue, which then becomes tacky, while HPC coating is masked from the UV light to avoid any potential degradation (Fig. 1d). The coated HPC film is then encapsulated using a R2R continuous laminator (Fig. 1e). The lamination gap (between the two PET substrates) is set to be 100 μm smaller than the HPC thickness with an aim to balance air entrapment and lateral flow that screens the adhesive. Adhesive bonding between the substrates is also triggered at this point by the nip pressure of the rollers during the lamination process. Throughout the R2R fabrication the HPC experiences various external forces

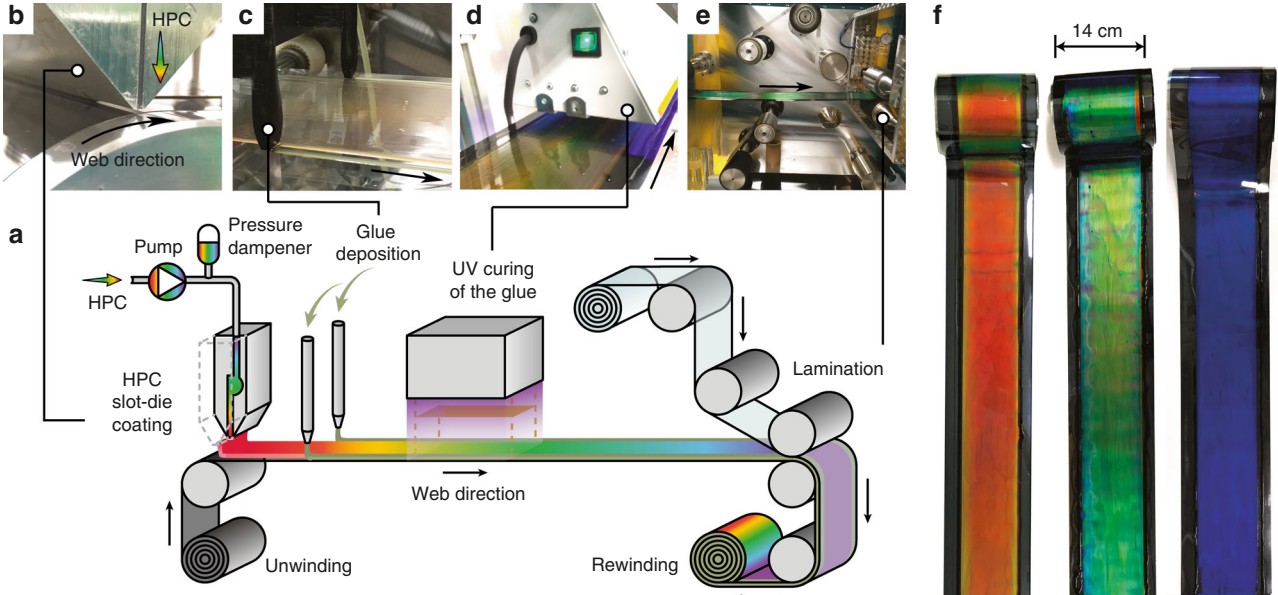

**Fig. 1** R2R fabrication of HPC-laminated films. **a** Schematic of the R2R fabrication for HPC-laminated films showing the sequential processes of: **b** slot-die coating of HPC; **c** edge sealing by glue deposition; **d** UV curing of the edge glue with mask shielding the HPC coating; **e** lamination for final packaging and rewinding. **f** Black PET-backed product rolls of red, green and blue HPC laminates with HPC concentrations of 63, 66 and 70 wt%, respectively

originating from the coating flow and web bending, deforming the cholesteric orientation from these influences at the molecular level. Consequently, the fully laminated mesophase requires additional relaxation of the residual stress in order to reach a stabilised colour (Supplementary Figure 5). The exact time required is concentration dependent, but in general the colour becomes visually consistent after approximately half an hour in all laminates produced.

**Colour mapping of HPC pressure response**. The above R2R coating experiments showcase the ability to continuously process self-assembled HPC into cost-effective, large area mechanochromic sensors. In order for these to be useful for practical applications, we calibrate the pressure response of the HPC mesophase by measuring their colour response under a range of pressures using two complementary methods: (1) a spectrometer to correlate the spectral response to the applied pressure in a way that is representative of the human eye's perception of colours; (2) a digital camera (webcam) for recording two-dimensional pressure maps. As depicted in Fig. 2a, the HPC sample is contained between a rigid glass plate and black PET film with an 800 µm spacer. Pressure is applied by pushing on the sample with a finger while the loaded force is measured using a calibrated capacitive force sensor (Supplementary Figure 6). This set-up allows for simultaneous measurements of colour and pressure changes (data processing is shown in Supplementary Figures 7–8). A resulting colour-shift covering the full visible spectrum is obtained within a pressure range from 0 to 10 kPa.

For spectral measurements, we use an integrating sphere to illuminate the HPC sample with diffuse light, while the reflection spectrum is collected at normal incidence (Fig. 2b). The lighting conditions correspond to common environmental lighting conditions and accounts for the scattering properties of the HPC mesophase, which reflects light from multiple incident angles at the same reflected angle due to the presence of tilted domains in the material. Colour changes from reflection spectra of compressed HPC are calculated in CIELAB colour space and plotted as 2D polar representation on the $a^*$ (red-green) $b^*$ (blue-

yellow) plane (Fig. 2c), where $C^* = \sqrt{a^{*2} + b^{*2}}$ (chroma, accounting for saturation), $h° = \arctan(b^*/a^*)$ (hue, as an angle on the colour wheel)[36,37]. This method is based on two standard configurations (ISO 13655 and ASTM E2539-12) adapted to match more closely the viewing conditions of a human observer.

For the camera measurements, the webcam faces the HPC sample at normal incidence under diffuse illumination (Fig. 2d). Videos are recoded during pressure loading. An area of $5 \times 5$ mm$^2$ at the centre of the colour changing region is used for measurement, as it has dimensions similar to the iris in the spectral set-up (see Methods). Averages of each R, G and B channel of all pixels within this region are extracted as the mean RGB colour, which is then converted to $H$ (hue), $S$ (saturation), $L$ (lightness) values. Colour coordinates are plotted in a ternary diagram with three primary colour axes R, G and B (Fig. 2e, detailed in Supplementary Figures 9, 10), where for a given point nearer to a vertex (R, G or B) the more substantial the colour is in the composition. The hue $H$ of a point can be viewed as its projection on the RGB triangle.

The CIELAB colour space represents in three coordinates every colour that is perceivable by human eyes: it contains colour coordinates of any spectrum within the visible range (from 380 to 780 nm), calculated within the assumption that the colours they represent are viewed by a standard observer (CIE 1931 2°) and lit by a standard illuminant (Daylight D65). The CIELAB colour space is perpetually uniform (the Euclidean distance between any two colour points equals to first approximation the sensitivity colour difference to the human eye) and the hue $h°$ represents the sensation of hue change on the rainbow wheel[37]. The RGB colour system is a digital tool to record colours by splitting them into intensities on three primary channels (R, G and B) with no assumption on the conditions in which the physical coloured object is illuminated. The maximum recordable intensities depend on the capacity of the imaging device. The hue $H$ accounts for the shifts of the colour input from one channel to another, therefore representing a shift on the rainbow wheel[36,38]. The applied pressures are divided into groups of 1 kPa increment. Colour charts obtained from both spectral and camera data show clear trends: the hue shifts from red at low pressure (1–3 kPa) to

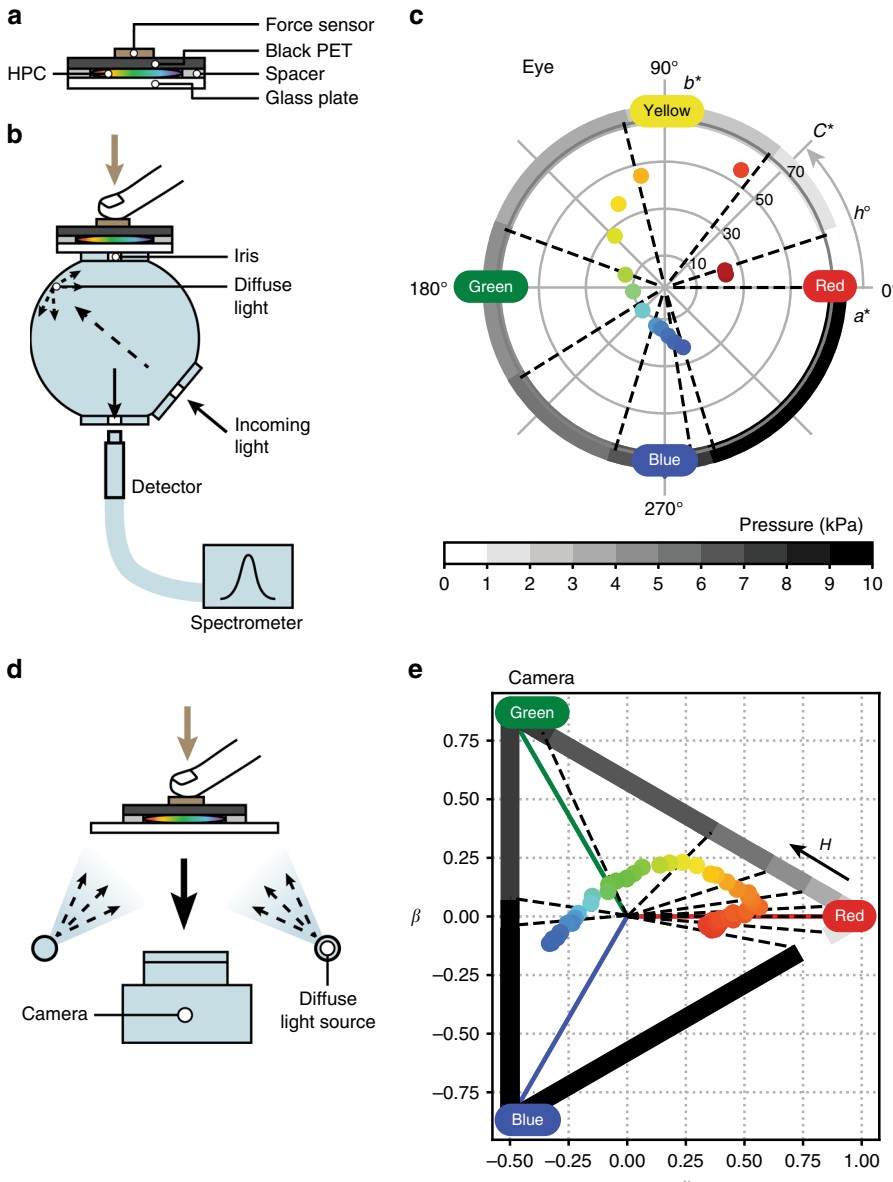

**Fig. 2** Measurements of pressure-induced colour change in HPC films with data plotted in CIELAB and RGB colour diagrams. **a** Sample HPC encapsulated between rigid glass plate and sheet of black PET backing, with 800 μm-thick spacers. **b** Set-up to measure reflection spectrum of HPC under pressure using integrating sphere with diffuse illumination and normal angle detection. Black arrows indicate light paths, iris diameter is 5 mm. **c** Colour diagram on $a*b*$ plane showing colour response of HPC under a pressure sweep. Axis $a*$ goes from green ($-a*$) to red ($+a*$), axis $b*$ goes from blue ($-b*$) to yellow ($+b*$). In polar coordinates, each point is defined by its radius to the centre (chroma $C*$) and its angle (hue $h°$). Hue $h°$ is mapped to applied pressure, pressure increments represented in shades of grey. **d** Set-up to capture colour of HPC under pressure, lit by diffuse light and with imaging device recording at normal angle. Black arrows indicate light path. **e** Ternary plot showing colour signal variation of HPC under a pressure sweep. Each point is defined by the relative weights of the R, G and B primaries (the vertices of the triangle). The hue $H$ is mapped to applied pressure, pressure increments represented in shades of grey on same scale as **c**

green at intermediate pressure (3–5 kPa), and onto blue under elevated pressure (5–10 kPa). At 0 kPa, the recorded spectrum is at the infrared boundary, which explains the unsaturated starting point in the data.

The observed increase in hue ($h°$ and $H$) with pressure corresponds to the shift towards shorter wavelengths in the reflection spectrum, which arises from the shortened average cholesteric pitch under pressure. However, at the microscale, vertical compression of locally tilted domains also results in pitches which are either smaller or larger depending on their relative helix orientation to the vertical compression, leading to localised relative blueshifts or redshifts. In addition, lateral shear

flow induced from pressure loads can also encourage domain misalignment (depicted in Fig. 3c), which leads to a broadening of the reflection spectrum, as evident in Fig. 3a. The result for colour perception is a wider spectral distribution of reflected wavelengths, perceived as a less saturated colour. These blue-shifting and desaturation effects are visible both in the spectrometer and camera data, showing the direct consequence of applying pressure on the HPC mesophase and how this affects the correlated human-perceptible and camera-perceptible colour.

Using optical microscopy, small areas of different colours are observed in the compressed HPC mesophase (Fig. 3b), evidencing the previously noted phenomena. Accounting for the resolution

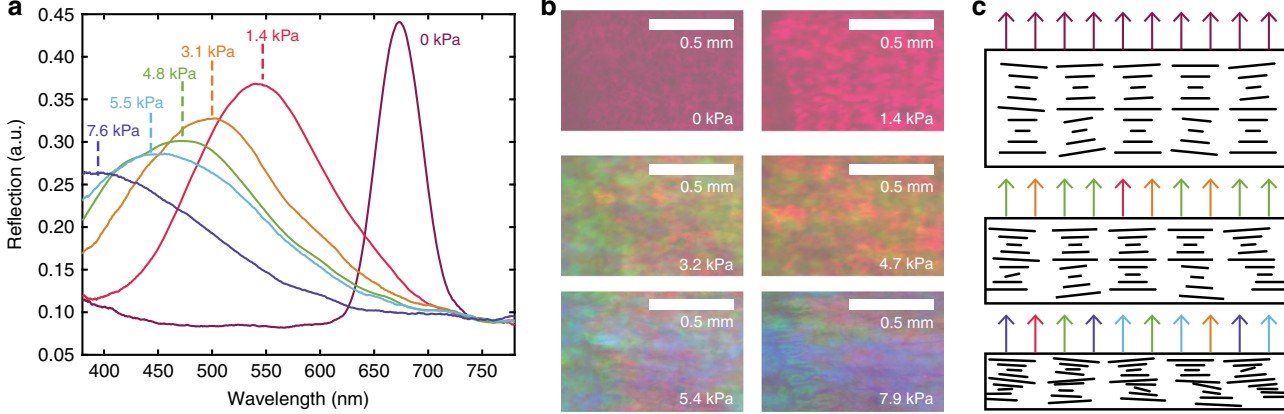

**Fig. 3** Microscopic colour variations resulting from local mis-aligned HPC helicoids under pressure loads. Reflection spectra (**a**) and optical microscope images (**b**) of sample HPC under continuous finger pressing. **c** Schematic representation of HPC domains becoming more disordered on compression. Depending on helix orientation with respect to force normal, pitch is either shortened or expanded locally, causing local colour deviations towards the blue or red in corresponding microscopic images and broadening the spectra in **a**

limit of the human eye, a viewing distance below 35 cm is required to resolve any difference of colour between tilted domains. At greater distances, such local colour imperfections merge and the human eye does not distinguish such microscopic variations. This complements the use of our HPC mesophase laminates as large-scale pressure sensors, where changes in colour are viewed and captured at typically longer distances, visually representing the distribution of applied pressure.

**Mechano-chromic pressure map of footprints on R2R HPC laminates**. The above demonstration of controlled large area flexible films along with our mobile phone-based pressure mapping paves the way for a myriad application ranging from sporting apparel to medical imaging. As an example, we map the pressure resulting from a foot imprint in real-time using a smartphone video. The average RGB colour is calculated from squares of $10 \times 10$ pixels (to minimise effects of camera noise and small pixel-to-pixel variations) and converted to the corresponding $H$ value. By nominal calibration of $H$ to the corresponding pressures between 0 and 10 kPa as described above, the foot pressure distribution can be plotted in false colours as shown in Fig. 4 lower panel. The real-time video allows for frame-by-frame extraction to obtain the dynamics and motion signature of a foot imprint (Supplementary Movie 1). As an example, we observe that the subject places little pressure on their small toe as the colour change of this region is absent throughout the recording. Overall, this demonstrator shows the ability to read-out 2D pressure maps over time using a low cost mechano-chromic HPC film and a mobile phone camera rather than arrays of pressure sensors which would be substantially more expensive and complicated to read out and analyse. Spatial resolution can be controlled through viscosity, thickness, and molecular weight, but already reaches ≪ 1 cm as required for insole plantar applications[39]. We therefore believe these films will be of interest for this application, as well as other industrial, medical, and entertainment applications. For each of these applications, the active material, processing parameters, and lamination materials will need to be optimised to meet specification that are determined by the application, such as the sensing resolution and lifetime requirements. We anticipate that a wider exploration of different HPC molecular weights or mixtures of materials, as well as modifications in the coating and encapsulation methods will allow to target specific sensing applications.

Overall, self-assembly of large molecules and nanoparticles allows for the development of materials with novel properties. To date however, most self-assembled structures are demonstrated on very small areas under well-controlled lab environments. Here, we show a R2R slot-die coating and lamination process, that allows for the continuous self-assembly of HPC into mechano-chromic materials over square meter areas. We find that by controlling the initial HPC solution and coating parameters, we can obtain films with arbitrary base colours across the visible spectrum. We also show how the mechano-chromic pressure response of these films can be calibrated, which in turn allows the recording of 2D pressure maps using standard mobile phone cameras. We demonstrate that this approach enables the recording of pressure profiles generated by foot-imprints in real time, and therefore this work paves the way for many applications requiring cost-effective large-area mapping of pressure distributions.

## Methods

**Materials**. PET rolls (PMX727-clear, PMX290-black, 75 μm thick, 140 mm wide) were purchased from HiFi Industrial Film, Stevenage, UK. UV curable pressure sensitive adhesive (UV PSA, Kiwoprint UV92) was purchased from The Screen Machine Supply, Milton Keynes, UK. HPC is commercially available (Nisso Chemical). HPC and water (Milli-Q) of designated weight ratios were mixed in a planetary mixer (Cuisinart SM50U) for 30 min for each 500 g batch. To ensure a bubble-free coating, degassing of the mixture was obtained by high speed centrifugation (14K rpm in Sorvall RC-5C) over a minimum of 2 h.

**Slot-die coating**. The slot-die is made of two screw-joined aluminium plates that are spaced by a stack of eight laser-cut pieces of 125 μm-thick PET film, which is configured for a 1 mm slot clearance. The supply of HPC is from a peristaltic feed-pump to the slot-die through a pressure dampener (Cole-Parmer Masterflex). The flow rate is set at 7.5 ml min$^{-1}$. The coating gap (distance between the slot-die lip and the substrate) was set to 1.2 mm. The source bottle (ThermoFisher Nalgene PPCO), pressure dampener and pumping tube are made of polypropylene, PTFE and silicon rubber, respectively, which are all low water vapour permeability materials.

**Edge sealing adhesive deposition and curing**. Two stripes of 10 mm wide UV PSA are deposited through 3D printed nozzle dispensers using a syringe pump at a flow rate of 0.8 ml min$^{-1}$. The nozzles are positioned for laydown the adhesive adjacent to the two edges of the coated mesophase film. The adhesive then passed under a fluorescent UV lamp (20 W, 365 nm) and was partially cured. This curing effect can be observed as a colour change from yellow to white cloudy (or clear) and an increase in tackiness. The HPC mesophase area was shielded from UV exposure by a Perspex mask covered by UV blocking film Amber 81.

 **5**

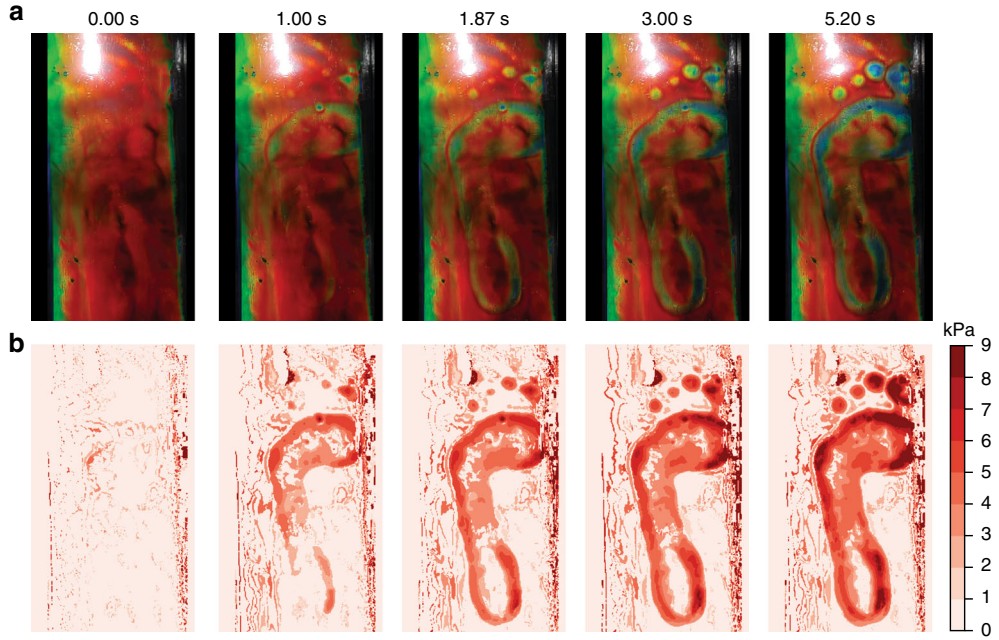

**Fig. 4** Frame-by-frame extraction of a footprint recorded on R2R red HPC laminate (14 cm width) from a 9-year-old participant. **a** Images are pixelated into 10 × 10 superpixels, with extracted average RGB values converted to $H$. **b** False-colour pressure maps of the footprint obtained via nominal calibration of $H$ by pressures retrieved from measurements in Fig. 2d, e. Full video of footprint recording is in the Supplementary Movie 1

**Lamination**. The lamination gap is determined by stainless feeler gauges with an initial setting of 850 μm considering the total substrate thickness. The gap is fine-tuned during actual operation. The final laminate is designed to be air entrainment-free without mesophase overflow affecting the sealing performance.

**Measurement of moisture loss during R2R processing**. Three liquid samples (6–7 g) of the mesophase are taken from each of the starting materials and their final laminates, and baked in a vacuum oven at 80 °C for 24 h. The loss of moisture was obtained by measuring the wt% change of the solid content.

**Encapsulated HPC samples used for colour mapping**. Spacers of 800 μm to maintain the thickness of HPC were fabricated by stacking up layers of poly-propylene and double-sided tapes attached to the long edges of a $50 \times 70$ mm$^2$ microscope glass plate. An HPC solution of the same concentration as in the R2R produced laminate was manually spread on the glass plate and encapsulated by laying down a sheet of black backing PET (75 μm), sealed by double-sided taped spacers. The sample was then left to rest for 3 h to allow the HPC to self-assemble.

**Measurement of RGB colour of HPC under pressure**. The HPC sample was secured above a webcam (Logitech c920) with two diffuse KL 1500 LCD light-sources illuminating from underneath. A force sensor (SingleTact 4.5N, Arduino compatible) was placed on top of black PET backing. Pressure was applied to the HPC by a finger loading onto the whole surface of the sensor, which was pro-grammed to read force values every 400 ms with each data input set to trigger the capture of a video frame by the camera. This method enabled a perfect colour–pressure match.

**Measurement of spectra of HPC under pressure**. A force sensor attached HPC sample was used as described above. The setup was placed on top of an integrating sphere (Labsphere) with an opening diameter of 5 mm. The illumination port of the integrating sphere (at 15°) was coupled to a 600 μm optical fibre connected to a xenon lamp (HPX-2000, Ocean Optics). The detector port of the integrating sphere was connected to a reflective collimator mounted on the detector and coupled to an optic fibre connected to a spectrometer (AvaSpec-HS2048, Avantes). The light intensity was normalised with respect to a white diffuser. The spectral measure-ments were performed by collecting light over 1 s for each spectrum, and auto-mated to record continuously on pressure loading.

**Optical microscope images**. Optical microscopy was performed using a Zeiss AxioScope optical inverted microscope equipped with a ×5 objective (Zeiss, LD EC Epiplan-Neofluar) and a CMOS camera. The above HPC sample setup was used with pressures applied as previously described. Screenshots were taken

automatically at each pressure input by the computer with the time and the force values stamped in the file name.

**Foot pressure–colour recording**. A R2R fabricated HPC laminate (from the red mesophase) using black and clear PET films as encapsulating substrates was placed on a clear Perspex plate with black PET as the top layer. The plate was suspended allowing optical access from underneath, while a human foot walked on top of the laminate. A camera (smartphone LG G4) from below recorded the colour change map of the laminate with a faster frame rate (30 fps).

**Code availability**. The computer code for image processing and colour space calculation is available on request addressed to S.V. (sv319@cam.ac.uk).

## Data availability
All relevant data are included in this article and its Supplementary Information files. Correspondence and additional requests are available from the corresponding author on reasonable request.

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

## Acknowledgements

This work is funded by EPSRC grants EP/N016920/1 to J.J.B., S.V., M.D.V. and H.-L.L.; EPSRC grant EP/L027151/1 to J.J.B and J.P.; by BBSRC David Phillips fellowship BB/K014617/1 to S.V.; by ERC-2014-STG H2020 639088 to S.V., R.V. and M.M.B.; C.H.B.-K. is supported by EPSRC EP/L016567/1; J.P. is supported by China Scholarship Council. H.-L.L. and M.M.B. thank Vanille Henry for help with the human dynamics, and Pierre-Henri François for useful discussions.

## Author contributions

H.-L.L., M.M.B., R.V., J.J.B., M.D.V. and S.V. conceived and designed the experiments. H.-L.L. and M.M.B. carried out the experiments, analysed and interpreted the data, and co-wrote the article with input from all authors. R.V. performed preliminary experiments, C.H.B.-K. performed the rheology measurements. J.P. involved in R2R experiments.

## Additional information

**Competing interests:** The authors declare no competing interests.

