## [Peer Review File · Nature Communications]

Reviewers' Comments:

Reviewer #1:

Remarks to the Author:

The authors present a R2R process to manufacture pressure-responsive films using the cellulose derivative hydroxypropyl-cellulose (HPC). Moreover, the HPC film applied to generate a real time pressure maps, which can find applications in sports apparel, medical imaging, strain monitoring and many other areas. The technology presented in this manuscript is attractive and innovative, and has the potential to be applied in many applications. The description of the fabrication method for realizing the special kind of nanopattern is clear. The reviewer suggests the manuscript be published after several minor revisions and corrections are made which help readers get a better understanding of this paper. The following comments could be considered in a revision;

1. Could you please provide a detailed description of the HPC used in the paper to help the reader reproduce? Now, there is only a description of mixing water and HPC powder.
2. Possibly to prevent the confusion of readers, fixing 88 lines of fig.S3 -> fig.S2, and 99 lines of fig.S2 -> fig.S3 could be done.
3. In fig.S2, HPC viscosity was measured at 22 degrees Celsius. Adding viscosity change data at lower or higher temperature is likely to be more reliable.
4. Regarding the slot-die coating method which minimizes water evaporation, quantitative measurement could be added (if possible) for better performance.
5. The reviewer believes the relaxation time may be different depending on the concentration of HPC coating. In Fig.S5, only green line is quantitatively measured. If the relaxation time of each color (concentration) could be measured, it could better verify that relaxation is smooth in the entire process.

Reviewer #2:

Remarks to the Author:

The authors have produced an excellent manuscript which meets a well-defined need in terms of discussing manufacturing technologies that could be applied in the fabrication of large area, flexible photonic materials.

The properties and potential applications of the resulting laminated photonic 'sheets' are ably demonstrated using pressure applied by a human foot, which is both informative and entertaining.

I am happy to recommend that the paper be published as is, with no corrections being required.

Reviewer #3:

Remarks to the Author:

The authors present a way to fabricate HPC based touch responsive laminates using a roll-to-roll method. The work demonstrated is of excellent quality, and there has been a good progress made in developing large area laminates. The manuscript, however, has some serious deficiencies which need to be addressed by the authors carefully.

- 1) First, although a roll-to-roll machine has been used to develop and demonstrate touch responsive sensors, neither the material used nor the process considered is novel or represent a scientific breakthrough. Yes, the authors have put in a lot of effort in customizing the system to develop the large area laminates, however, there isn't any science involved to warrant consideration by Nature Communications.
- 2) Nonuniformities are evident throughout the roll, as can be seen in Fig. 1f. Although blue looks uniform, red and green are not.
- 3) The authors need to explain why and how this kind of a pressure sensor is superior over the

current state-of-the-art. Given that this pressure sensor is applicable to niche applications (in contrast to what is being claimed by the authors), this discussion is extremely important. In other words, what is the figure of merit (FOM) and how does it compare with the current practice?

4) The different colors are obtained by controlling the evaporation rate. This in itself is a fine control that will slow the process down and very quickly negate the advantages of choosing a roll-to-roll process.

5) A device such as this, especially with a compressible gap, needs to be tested for usage cycles. Even if utmost care can be taken to achieve the precise colors, pressure nonuniformities that develop over time from usage or usage cycles can affect the performance. The authors need to consider performing extensive tests on the reliability of the laminates presented. If a hard surface such as glass or metal is needed as a background, its applications for athletics and others becomes difficult.

6) Since the devices are not exactly pixellated, severe crosstalk from adjoining pressure points will be coupled when detected by the camera. This does not present the actual pressure at that position, but a superposition of all pressure activity around that area.

7) The pressure maps need to be referenced against a baseline (a commercial pressure sensor) to account for differences. As is, the devices show that they can detect pressure changes, but it is not clear if the quantitative pressure is accurate.

8) Finally, the work seems to be more suitable for a manufacturing techniques related journal, and I would strongly encourage the authors to consider such journals where emphasis on process is more than the underlying science.

Response to Reviewers

Reviewer #1:

1. Could you please provide a detailed description of the HPC used in the paper to help the reader reproduce? Now, there is only a description of mixing water and HPC powder.

We have now added details of how they have been mixed with water, in the Methods section (Line 250 – 251): “HPC and water (Milli-Q) of designated weight ratios were mixed in a planetary mixer (Cuisinart SM50U) for 30 minutes for each 500g batch”, and we give the source of HPC in the original text.

2. Possibly to prevent the confusion of readers, fixing 88 lines of fig.S3 -> fig.S2, and 99 lines of fig.S2 -> fig.S3 could be done.

This potential source of confusion is fixed by adapting the manuscript as suggested.

3. In fig.S2, HPC viscosity was measured at 22 degrees Celsius. Adding viscosity change data at lower or higher temperature is likely to be more reliable.

If we understand the reviewer correctly, they are concerned about the effect of temperature variation on the viscosity. We are very aware of this effect and have measured our data specifically at 22°C because this is the temperature at which our coating experiments are conducted. The reported viscosity is therefore the one that is relevant for the coating performed in this work. We have clarified this in the manuscript (Line 88–89) and Fig. S2 caption.

4. Regarding the slot-die coating method which minimizes water evaporation, quantitative measurement could be added (if possible) for better performance.

As discussed in the manuscript, water loss of coated HPC mixtures was measured as 3 – 4 wt% over the coating to lamination process, which is equivalent to 6 – 8 wt% per hour (slightly dependent on the environmental conditions). This is too high for a continuously operating knife-over coating. However, during the slot-die coating process, the HPC is contained in a complete enclosed environment with low moisture permeable containers, hence lower water evaporation:

1. The main slot-die body is a >2cm thick water-impermeable aluminium block,
2. HPC is pumped from a capped polypropylene bottle through silicone rubber tubing to the slot-die. The water vapour permeability of polypropylene and silicone rubber are in the range of 0.6 and 3 g·mm/m²·day, respectively,
3. All fittings used are air- and water-tight, using push-in Festo, screw, and barbed fittings.

Because of this, no evaporative problems were observed during our slot die coating experiments. We add these clarifications in the methods section of the revised manuscript. (Line 254–261)

5. The reviewer believes the relaxation time may be different depending on the concentration of HPC coating. In Fig.S5, only green line is quantitatively measured. If the relaxation time of each color (concentration) could be measured, it could better verify that relaxation is smooth in the entire process.

The reviewer is correct that the relaxation time varies with HPC concentration (and molecular weight) and therefore differs for each colour even if the same material is used. Fig. S5 shows an example of the relaxation process after the mesophase-distortion induced by the shear applied during the manufacturing process. The effect was not quantified in detail but was evaluated visually until no apparent colour changes is detectable to the eye. We report this in the SI only to point out that there is a delay-time in which the roll needs to rest before it can be

used for measurements. In general, we did not observe distinguishable changes after 30 minutes resting in all the laminates that were produced, independent of their colour. In the SI, we show the green laminate example because the phenomenon is most clearly visible (since the human eye is most sensitive to green under normal illumination conditions). We now also explicitly mention in the main text that the relaxation time is concentration and molecular weight dependent. (Line 131-133)

Reviewer #3:

1. Although a roll-to-roll machine has been used to develop and demonstrate touch responsive sensors, neither the material used nor the process considered is novel or represent a scientific breakthrough. Yes, the authors have put in a lot of effort in customizing the system to develop the large area laminates, however, there isn't any science involved to warrant consideration by Nature Communications.

We disagree with the reviewer, as the realization of a *truly scalable* fabrication of photonic strain-sensors made out a sustainable and cheap material like HPC is worthy of the attention of the broad audience of *Nature Communications*. Our work is the first example where self-assembly is used in a R2R machine to process bio-compatible polymers into functional films. We have not only proven that the large-area production of these materials is possible, but also show a sensing application along with careful characterisation experiments. Further we have also developed a read-out system which can provide a direct pressure data from the HPC films.

2. Nonuniformities are evident throughout the roll, as can be seen in Fig. 1f. Although blue looks uniform, red and green are not.

To our understanding, the non-uniformities to which the reviewer refers are the areas showing colour variations in the laminate. The colour of the HPC comes from the dimensions of the helical pitch and/or the orientation of the helical domains in the mesophase and any changes of these two parameters results in a colour variation. In our laminates, defects originate from:

- a) moisture loss, visible at the edge of the film due to imperfect sealing: a blue-shift in colour is observable along the edge of the laminates, which can be addressed by coating wider laminates (ours are only 10 cm wide) and better sealing.
- b) bending stress from laminate handling, which gives a visible defect pattern characterised by periodic stripes across the film width: this effect can also be addressed using larger diameter rolls or by leaving the film at rest for a longer time.

Both of the above drive blue-shifts in colour and therefore the non-uniformities are easier to observe in red and green laminates than in blue (where the defects now reflect in the UV region and are not visible to the camera/eye).

We also clarify that the red laminate in Fig.1f was made from 60 wt% HPC, which actually reflects in the NIR-red region and therefore is a mix of invisible and visible colours. To show a large area uniform red is possible, we have reproduced a laminate from 63 wt% HPC and replace the NIR-red laminate with this in Fig.1f. It now shows a consistent red colour.

Original Fig 1f and caption:

f, Black PET-backed product rolls of red, green and blue HPC laminates with HPC concentrations of 60 wt%, 66 wt% and 70 wt% respectively

Revised Fig 1f and caption:

f, Black PET-backed product rolls of red, green and blue HPC laminates with HPC concentrations of 63 wt%, 66 wt% and 70 wt% respectively

We agree with the referee that, for practical usage, a uniform baseline is required for good sensor performance (which is how we select uniform areas for foot pressure mapping). The above issues are for future improvement (as they are very machine dependent) and fall outside of the scope of this work.

3. The authors need to explain why and how this kind of a pressure sensor is superior over the current state-of-the-art. Given that this pressure sensor is applicable to niche applications (in contrast to what is being claimed by the authors), this discussion is extremely important. In other words, what is the figure of merit (FOM) and how does it compare with the current practice?

As discussed in the manuscript an important innovative aspect of this work is the ability to map 2D pressure distributions using a *large area* and *cost-effective* sensor. Mapping pressure distributions using traditional sensors would require building large sensor arrays, and this

option would be costly if high special resolution is required – indeed none are available currently. The material cost in our material is less than $<£10/m^2$. Because our sensing technology is so different from any existing solution, and because the main merits are based on the cost-effective manufacture of large area sensors, it is difficult to benchmark it against existing solutions. An additional advantage difficult to capture in a FOM is that our material is environmentally benign. A further advantage is that it is capable of detecting different deformation modes such as compression and shear, as discussed elsewhere (ref [26]).

4. The different colors are obtained by controlling the evaporation rate. This in itself is a fine control that will slow the process down and very quickly negate the advantages of choosing a roll-to-roll process.

The referee helpfully points out a potential confusion here. The different laminate baseline colours are actually obtained by controlling the initial concentration of the HPC source solution, rather than by controlling the evaporation rate in-line. Therefore they are not affected by the coating speed in our R2R process. We do carefully mention that the loss of moisture during the R2R process should be taken into account while setting this initial concentration. In our case, we measure a ~3% of water loss during the process and therefore accordingly slightly dilute the source HPC solution. This is now further clarified in our manuscript. (Line 66, 83–84, and 86–87)

5. A device such as this, especially with a compressible gap, needs to be tested for usage cycles. Even if utmost care can be taken to achieve the precise colors, pressure nonuniformities that develop over time from usage or usage cycles can affect the performance. The authors need to consider performing extensive tests on the reliability of the laminates presented. If a hard surface such as glass or metal is needed as a background, it's applications for athletics and others becomes difficult.

We certainly agree that long term fatigue and abuse testing are needed before this laminate material can be taken to market. However, we do not intend this scientific paper to present a retail-ready product which requires industrial fatigue testing capabilities outside the scope of this work. In fact this would mainly test the durability of the encapsulating lamination films which we could change depending on the intended use. Importantly, the underlying material operation is simply based on the change of pitch in the self-assembled HPC, and there is no evidence that this process is susceptible to fatigue. To better clarify this to the reader, we add an additional paragraph discussing these points. (Line 224–230)

6. Since the devices are not exactly pixellated, severe crosstalk from adjoining pressure points will be coupled when detected by the camera. This does not present the actual pressure at that position, but a superposition of all pressure activity around that area.

Pressure mapping devices vary in sensor configuration to meet different application requirements, and do not necessarily require pixilation. Insole plantar pressure sensors for example, often use 15 individual sensors to cover the major areas that support body weight based on foot anatomy. For this application, a minimum measurement area of 5 mm x 5 mm must be used [see eg. Sensors 12, 9884-9912 (2012)], and the 15 areas are on average 3cm apart. Although, our sensors are not pixelated, our measurements do not show any cross talk over such distances as exemplified by the colour changes from finger pressure we show in Fig. S8. Our colour calibration is performed across such a 5 mm x 5 mm area to match this reference. We revised the manuscript to address this relevant concern and added the above reference. (Line 221–223)

7. The pressure maps need to be referenced against a baseline (a commercial pressure sensor) to account for differences. As is, the devices show that they can detect pressure changes, but it is not clear if the quantitative pressure is accurate.

We have indeed performed this colour mapping and calibrated the colour response to pressure on small area samples using a commercial pressure sensor as described in the section ‘Colour mapping of HPC pressure response’ and in Fig.2. The referee is correct that this such quantitative connection is vital, hence the extensive efforts in our manuscript. For referencing large-area samples in a similar way requires a foot-size thin-film pressure sensor, a rigid pressure mat, or multiple small pressure sensor placements which are cross-calibrated, all of which are non-trivial. However the calibration provided is robust and quantitative as requested.

8. Finally, the work seems to be more suitable for a manufacturing techniques related journal, and I would strongly encourage the authors to consider such journals where emphasis on process is more than the underlying science.

We stress the several scientific challenges tackled here which show how nano self-assembly can be truly scaled to develop a continuous large-scale coating process. Understanding these challenges is particularly important to support the application of nanomaterials and self-assembly in real-life devices. We believe that the first application of large area 2D photonics will inspire other applications in the future and challenge the way people use bottom-up self-assembly in manufacturing. Therefore, we believe that this work well serves the wider audience of *Nature Communications*.

Reviewers' Comments:

Reviewer #1:

Remarks to the Author:

The manuscript is now ready for publication.

Reviewer #3:

Remarks to the Author:

The authors have addressed all of the concerns satisfactorily. This can be published as is in the Journal.